# Proteomics and *C9orf72* neuropathology identify ribosomes as poly-GR/PR interactors driving toxicity

Hannelore Hartmann[1,*], Daniel Hornburg[2,*] , Mareike Czuppa[1], Jakob Bader[2] , Meike Michaelsen[1], Daniel Farny[1], Thomas Arzberger[1,3,4], Matthias Mann[2,4], Felix Meissner[2], Dieter Edbauer[1,5,6]

**Frontotemporal dementia and amyotrophic lateral sclerosis patients with *C9orf72* mutation show cytoplasmic poly-GR and poly-PR aggregates. Short poly-(Gly-Arg) and poly-(Pro-Arg) (poly-GR/PR) repeats localizing to the nucleolus are toxic in various model systems, but no interactors have been validated in patients. Here, the neuronal interactomes of cytoplasmic GFP-(GR)₄₉ and nucleolar (PR)₁₇₅-GFP revealed overlapping RNA-binding proteins, including components of stress granules, nucleoli, and ribosomes. Overexpressing the poly-GR/PR interactors STAU1/2 and YBX1 caused cytoplasmic aggregation of poly-GR/PR in large stress granule–like structures, whereas NPM1 recruited poly-GR into the nucleolus. Poly-PR expression reduced ribosome levels and translation consistent with reduction of synaptic proteins detected by proteomics. Surprisingly, truncated GFP-(GR)₅₃, but not GFP-(GR)₄₉, localized to the nucleolus and reduced ribosome levels and translation similar to poly-PR, suggesting that impaired ribosome biogenesis may be driving the acute toxicity observed in vitro. In patients, only ribosomes and STAU2 co-aggregated with poly-GR/PR. Partial sequestration of ribosomes may chronically impair protein synthesis even in the absence of nucleolar localization and contribute to pathogenesis.**

## Introduction

Since the discovery of the $(GGGGCC)_n$ repeat expansion in *C9orf72* in about 10% of amyotrophic lateral sclerosis (ALS) and frontotemporal dementia patients, several potential pathomechanisms have been proposed (Edbauer & Haass, 2016). The repeat RNA is clustered in nuclear foci in neurons and non-neuronal tissues in patients, without apparent correlation with neuron loss (DeJesus-Hernandez et al, 2017). Several proteins binding to the repeat RNA have been identified, but up to now their role in pathogenesis is still

unclear. Furthermore, sense and antisense transcripts of the repeat are translated into five dipeptide repeat (DPR) proteins that co-aggregate in predominantly neuronal cytoplasmic inclusions in *C9orf72* patients: poly-GA, poly-GP, poly-GR, poly-PR, and poly-PA. Although several groups failed to detect a direct correlation of DPR expression with neurodegeneration (Mackenzie et al, 2013, 2015; Schludi et al, 2015), a recent report identified dendritic poly-GR pathology specifically in the motor cortex of ALS patients (Saberi et al, 2018), although it is unclear how the conflicting findings can be explained. Therefore, the role of the DPR proteins in disease pathogenesis is still under intense debate.

Individual expression of poly-(Gly-Arg) and poly-(Pro-Arg) (poly-GR/PR) is highly toxic in various model systems (Kwon et al, 2014; Mizielinska et al, 2014; Wen et al, 2014; Jovicic et al, 2015; Boeynaems et al, 2016; Lee et al, 2016; Lin et al, 2016; Lopez-Gonzalez et al, 2016), but poly-GR and especially poly-PR show predominantly nucleolar localization in most in vitro systems, which is not observed in patient tissue (Schludi et al, 2015). Genetic screens for poly-PR toxicity have highlighted a link to nucleocytoplasmic transport (Jovicic et al, 2015; Boeynaems et al, 2016), whereas poly-GR seems to predominantly affect other pathways in yeast (Chai & Gitler, 2018). Recently, poly-GR/PR have been shown to undergo phase separation in vitro and interact with low-complexity domain proteins in membrane-less organelles, specifically in the nucleolus and stress granules (Lee et al, 2016; Lin et al, 2016; Boeynaems et al, 2017). Poly-GR/PR–interacting proteins have been analyzed using pull-down experiments with short peptides that spontaneously phase-separate together with proteins and RNA from the cell extracts (Kanekura et al, 2016; Lin et al, 2016; Boeynaems et al, 2017; Yin et al, 2017) or using expression of short repeat constructs (Lee et al, 2016; Lopez-Gonzalez et al, 2016). In contrast to several poly-GA–interacting proteins (May et al, 2014; Zhang et al, 2016; Schludi et al, 2017), none of the reported poly-GR/PR interactors has been validated in patient tissue. Poly-GR/PR toxicity has also been linked to altered splicing (Kwon et al, 2014; Yin et al, 2017), reduced translation (Kanekura et al, 2016), ER stress (Kramer et al, 2018), and

[1]German Center for Neurodegenerative Diseases (DZNE), Munich, Germany    [2]Max Planck Institute for Biochemistry, Martinsried, Germany    [3]Center for Neuropathology and Prion Research, Ludwig-Maximilians-University Munich, Munich, Germany    [4]Department of Psychiatry and Psychotherapy, Ludwig-Maximilians-University Munich, Munich, Germany    [5]Ludwig-Maximilians-University Munich, Munich, Germany    [6]Munich Cluster of Systems Neurology (SyNergy), Munich, Germany

Correspondence: dieter.edbauer@dzne.de
*Hannelore Hartmann and Daniel Hornburg contributed equally to this work.
Daniel Hornburg's present address is Stanford University, School of Medicine, Palo Alto, CA, USA.

mitochondrial stress (Lopez-Gonzalez et al, 2016), but it is unclear which effects are relevant in patients. The severe toxicity in some model systems is hard to reconcile with the prodromal expression at least of poly-GR many years before disease onset (Vatsavayai et al, 2016). Therefore, current models likely exaggerate toxicity although it is possible that cytoplasmic poly-GR/PR inclusions trigger similar pathways in vivo with milder effects.

To elucidate the functional consequences of poly-GR/PR expression in patients, we analyzed the interactomes of poly-GR and poly-PR in primary neurons and HEK293 cells and validated candidate proteins in cellular systems and patient tissue, focusing on stress granules, the nucleolus, and ribosomes. Overexpression of several interactors recruits poly-GR/PR into large cytoplasmic stress granule–like structures. Moreover, acute neurotoxicity of poly-GR/PR requires nucleolar localization and is associated with reduced levels of ribosomes and impaired translation. Importantly, we could validate co-aggregation of ribosomes in cytosolic DPR inclusions in patient brain tissue, supporting a primary role of translational inhibition for poly-GR/PR toxicity in vivo.

## Results

### Poly-GR and poly-PR interact with ribosomes, stress granules, and low-complexity proteins

To identify which poly-GR and poly-PR interactors would be most relevant for neurodegeneration in *C9orf72* patients, we analyzed the poly-GR/PR interactomes in rat primary cortical neurons and HEK293 cells. Consistent with previous results (Schludi et al, 2015), lentiviral expression with GFP-(GR)$_{149}$ in neurons resulted in predominantly diffuse cytoplasmic expression and some nucleolar localization, whereas (PR)$_{175}$-GFP was mostly localized to the nucleolus (Fig S1A and B). Fusion with nuclear export signals or (GA)$_{50}$ failed to shift poly-PR quantitatively to the cytoplasm (data not shown). In HEK293 cells, (PR)$_{175}$-GFP also mainly localized to the nucleolus, whereas GFP-GFP-(GR)$_{149}$ was found in both nucleolus and cytoplasm. In contrast to previous reports, only (PR)$_{175}$-GFP, but not GFP-(GR)$_{149}$, induced significant cell death in neurons compared with the GFP control as measured by an LDH release assay (Fig S1C). However, both (PR)$_{175}$-GFP and GFP-(GR)$_{149}$ impaired the growth of HEK293 cells as shown by the XTT assay, which measures mitochondrial activity (Fig S1D) but did not trigger significant cell death as reported previously (May et al, 2014). Overall, these observations suggest that nucleolar localization may be important for poly-GR/PR toxicity in vitro.

For the interactome analysis from primary neurons and HEK293 cells, we immunoprecipitated GFP-(GR)$_{149}$, (PR)$_{175}$-GFP, and a GFP control using GFP antibodies and analyzed the interactome using quantitative mass spectrometry. In primary neurons, we quantitatively compared close to 600 proteins (Table S1A). Among those, we identified 89 poly-GR and 104 poly-PR interactors (Fig S2A), of which ~60% are annotated as RNA-binding proteins (Gerstberger et al, 2014). Both DPR proteins interact with numerous components of ribosomes, the nucleolus, and stress granules (Jain et al, 2016) as well as proteins involved in splicing. 39 proteins were commonly

enriched in both interactomes (Fig 1A). Consistent with previous data, sequence analysis of poly-GR/PR interactors shows enrichment of proteins with low-complexity domains (Fig 1B).

For comparison, we additionally analyzed the poly-GR/PR interactome from HEK293 cells, resulting in 394 proteins enriched exclusively in poly-GR and 49 proteins enriched in both poly-GR and poly-PR (Fig S2B and Table S1B). Only one protein (CD2AP) was solely enriched in the poly-PR interactome. In total, about 80% of the interactors are annotated as RNA-binding proteins (Gerstberger et al, 2014). Overall, there was a consistent overlap with published data (Lee et al, 2016; Lin et al, 2016; Boeynaems et al, 2017).

Comparison of the gene ontology (GO) terms enriched in the poly-GR/PR interactors in primary neurons and HEK293 cells showed a strong selectivity for proteins related to ribosomes, stress granules, the nucleolus, the spliceosome, and the methylosome (mediating arginine methylation) in the poly-GR interactome (Fig 1B and Table S2). Interaction of PRMT1/5 with poly-GR, but not poly-PR, suggests that only poly-GR is arginine methylated (Schludi et al, 2015). Poly-PR interactors were most strongly enriched in proteins of the U1 and U4 small nuclear RNP, the exon–exon junction complex and mitochondrial ribosomes in both cell types. Although several cytosolic ribosomal proteins were enriched in the (PR)$_{175}$-GFP immunoprecipitates in neurons (Fig S2A), several other subunits were depleted, which may be explained by the overall reduction of cytosolic ribosomal proteins in poly-PR–expressing neurons, whereas mitochondrial ribosomal proteins are even found at higher levels (Figs 5B and S6B).

In summary, poly-GR/PR interact preferentially with RNA-binding proteins. For the functional analysis, we focused on the interaction of poly-GR/PR with the nucleolus, stress granules, and the ribosome and compared findings from overexpression of several interactors in cultured cells with patient tissue.

### NPM1 traps poly-GR into the nucleolus

To functionally validate putative poly-GR/PR–interacting proteins and get a better understanding of their role in DPR toxicity, we co-expressed RFP-tagged interactors together with GFP-(GR)$_{149}$, (PR)$_{175}$-GFP, or control GFP in HEK293 cells and neurons and analyzed the localization of the DPR proteins and the interacting proteins.

First, we tested the nucleolar proteins NOP56 and NPM1, which we identified as interactors in both cell types (Fig 1A and Table S1). As expected, RFP-NOP56 and RFP-NPM1 co-localized with poly-GR and poly-PR predominantly in the nucleolus in HEK293 cells (Fig S3). In primary neurons, GFP-(GR)$_{149}$ was largely absent from the nucleolus (Fig 2A). However, expression of RFP-NPM1 surprisingly recruited cytoplasmic GFP-(GR)$_{149}$ into the nucleolus resulting in co-localization in almost all cells (Fig 2B and D). In contrast, poly-GR localized predominantly to the cytosol in RFP-NOP56–transduced neurons, similar to the RFP control (Fig 2A, C, and D). Poly-PR co-localized with RFP-NOP56, but co-expression altered the distribution of RFP-NOP56 within the nucleolus compared with GFP or GFP-(GR)$_{149}$ (Fig 2C, close-up in right column). Although RFP-NOP56 is evenly distributed in the nucleolus under control conditions in neurons, it showed a granular pattern in poly-PR–expressing cells, which is consistent with the poly-PR–specific interaction of NOP56

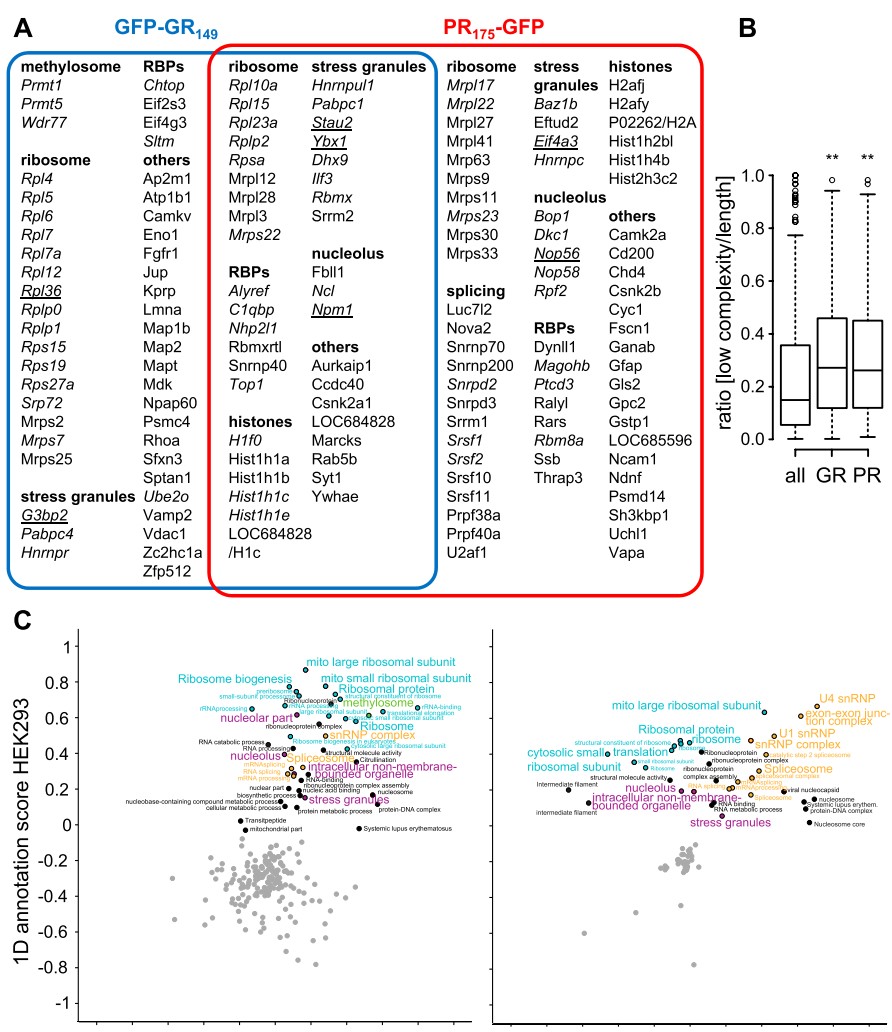

**Figure 1. Poly-GR and poly-PR interact with similar low-complexity proteins in neurons.**

Quantitative proteomics of GFP immunoprecipitations from primary cortical neurons transduced with GFP, GFP-(GR)$_{149}$, or (PR)$_{175}$-GFP (DIV7 + 8). The complete dataset is available in Table S1. **(A)** Proteins with significant enrichment in poly-GR/PR interactomes compared with GFP control were manually grouped into functional categories. Orthologues of proteins in italics were also found in the poly-GR/PR interactomes from HEK293 cells. Underlined proteins are analyzed in this paper. **(B)** Proportion of low-complexity regions (IUPred-L) of all proteins identified in the neuronal interactome analysis, the poly-GR interactome, and the poly-GR interactome. Significance of difference was assessed with the Mann–Whitney–Wilcoxon test, exact P-values: GFP versus GFP-(GR)$_{149}$, P = 0.004566 and GFP versus (PR)$_{175}$-GFP, P = 0.001656. Whiskers extend to ±1.5 box height (i.e., total three times the interquartile range). **(C)** 2D analysis of GO enrichment terms (GOMF, GOCC, GOCC, GOPB, KEGG, and UniProt keywords) and stress granule proteins (Jain et al, 2016) for proteins found in the poly-GR and poly-PR interactome in primary neurons and HEK293 cells (Fig S2 and Tables S1 and S2). Some dots with nearly identical position and annotation were removed for clarity. Related terms from the main enriched pathways are labeled in the same color. Annotation terms with a Benjamini–Hochberg FDR (q-value) <0.1 and comprising at least six proteins quantified by mass spectrometry are shown. 1D annotation scores close to 1 indicate strongest enrichment over the GFP control, scores close to 0 indicate no enrichment, and scores close to −1 indicate strongest depletion. The analysis was performed in Perseus software (Tyanova et al, 2016).

in neurons (Fig 1A). Thus, the interaction of poly-GR/PR with nucleolar proteins has sufficient affinity to alter the subcellular distribution of either binding partner.

## STAU1/2 and YBX1 reroute poly-GR/PR into large cytoplasmic granules

The poly-GR/PR interactomes contain many stress granule–related proteins, but only a small fraction of transfected cells shows small cytoplasmic poly-GR/PR granules (arrows in Fig S3A). However, expression of several RNA-binding proteins from the poly-GR/PR interactome promoted cytoplasmic clustering of the two DPR protein species in HEK293 cells. Most strikingly, STAU1/2 and YBX1 rerouted both poly-GR and poly-PR into cytoplasmic clusters that can be quite large (Fig 3A) and are reminiscent of the cytoplasmic poly-GR/PR inclusions seen in patients (Mori et al, 2013a, b). In contrast, we did not detect any differences with the stress granule–associated poly-PR interactor EIF4A3 (Fig 3A and B). Quantitative analysis showed that the average size and also the

number of cytoplasmic poly-GR/PR inclusions are strongly increased upon expression of RFP-STAU1/2 and RFP-YBX1 but not RFP-EIF4A3 (Fig 3B).

Using transient co-transfection to allow higher expression levels in primary neurons, we detected similar co-localization of GFP-(GR)$_{149}$ with RFP-STAU1/2 and RFP-YBX1 in cytoplasmic clusters (Fig 3C) but not for RFP-EIF4A3 (data not shown). Presumably because of even higher toxicity, we did not detect (PR)$_{175}$-GFP–expressing neurons upon transfection. Thus, the interaction with several stress granule–associated proteins recruits poly-GR/PR into large cytoplasmic granules resembling the neuronal cytoplasmic inclusions seen in *C9orf72* patients.

## Cytoplasmic poly-GR/PR clusters resemble stress granules

To elucidate the nature of these cytoplasmic poly-GR/PR clusters, we probed HEK293 cells co-transfected with RFP-STAU1 and poly-GR/PR with the stress granule marker G3BP1 and detected striking co-localization and sequestration of G3BP1 into the poly-GR/PR granules (Fig 4A). Without co-expression of RNA-binding proteins,

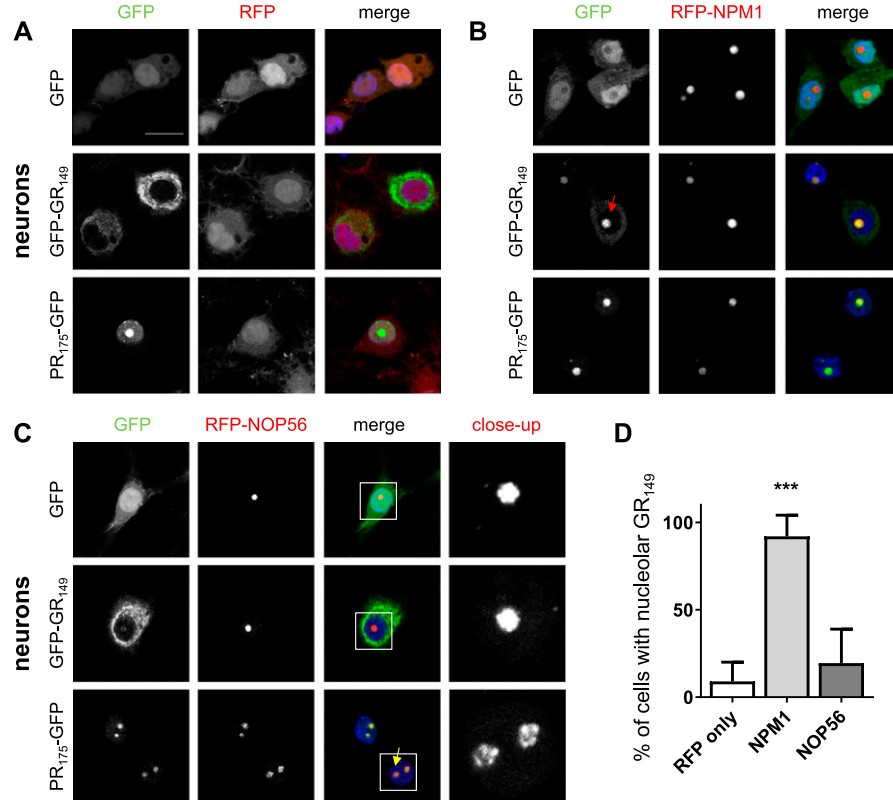

**Figure 2. NPM1 drives poly-GR into the nucleolus of primary neurons.**
Primary rat neurons (DIV7 + 7) were cotransduced with RFP-tagged nucleolar interactors NPM1 and NOP56 with GFP, GFP-(GR)$_{149}$, or (PR)$_{175}$-GFP. **(A–C)** Single focal planes obtained by confocal microscopy are shown. DAPI was used as nuclear marker and the scale bar depicts 20 μm. RFP was used as negative control. Left columns show GFP signal, middle columns show RFP-tagged proteins, and right columns show merge of GFP, RFP-tagged proteins, and nuclear DAPI staining (blue). RFP-NPM1 and RFP-NOP56 are co-localizing with poly-PR. Note that NPM1 expression recruits poly-GR into the nucleolus (red arrow). Poly-PR–expressing neurons show altered NOP56 nucleolar staining (single channel shown in zoom). **(D)** Fraction of cells with poly-GR localized to the nucleolus in NPM1– and NOP56–expressing neurons compared with the RFP control (RFP, n = 9; RFP-NPM1, n = 11; and RFP-NOP56, n = 10 images (40× objective) from three independent experiments, mean ± SEM, exact *P*-values: RFP versus RFP-NPM1, *P* = 0.0001 and RFP versus RFP-NOP56, *P* = 0.2307 in one-way ANOVA with Dunnett's posttest).

the less frequent cytoplasmic poly-GR/PR punctae were predominantly G3BP1 positive, indicating that overexpression of STAU1/2 and YBX1 enhances a normal process that may ultimately lead to aggregation of poly-GR/PR in patients (compare Fig 3B), which is consistent with the interaction with stress granule proteins under basal conditions (Fig 1).

Then, we wondered whether poly-GR/PR inclusions in patients could be labeled by stress granule markers and compared an frontotemporal lobar degeneration case with *C9orf72* repeat expansion with a healthy control case by double immunofluorescence. As expected, the *C9orf72* frontotemporal lobar degeneration case showed widespread poly-GR and sparse poly-PR cytoplasmic inclusions in the frontal cortex. We detected not a single poly-GR/PR inclusion convincingly co-localizing with classical stress granule markers proteins (G3BP2 and TIAR) and the interactor YBX1 in two *C9orf72* patients. However, ~25 % of poly-GR inclusions (76 of 300 counted aggregates) were co-stained with STAU2 in cortex (Fig 4B).

In conclusion, despite interaction of poly-GR/PR with many stress granule–related proteins and recruitment of poly-GR/PR into stress granules on overexpression of STAU1/2 and YBX1, classical stress granule marker proteins are not readily detectable in the poly-GR inclusions in postmortem brains of *C9orf72* patients, suggesting a more transient interaction.

### Poly-GR/PR inclusions in patients contain ribosomes

Because the ribosomal proteins are very prominent in the poly-GR/PR interactomes, we additionally analyzed the localization of the 40S

protein RPS6 in poly-GR/STAU1–co-transfected HEK293 cells (Fig S4A). Unlike for stress granule markers, we detected only modest amounts of RPS6 in poly-GR/PR inclusions without strong enrichment compared with the cytoplasm.

We did not analyze individual ribosomal proteins using the co-expression approach because tagging the ribosome is notoriously difficult. Instead, we directly analyzed the localization of several ribosomal proteins with good available antibodies directly in patient brain. Compared with controls, several ribosomal subunits (RPS6, RPS25, RPL19, and RPL36A) were enriched in both poly-GR and, less strikingly, also in poly-PR inclusions (Figs 5A and S4B). Quantification shows that approximately one third of GR inclusions show co-localization with ribosomal proteins in cortex (Figs 5B and S4C). However, most neurons showed robust residual ribosome staining in the cytoplasm.

Moreover, we extensively tested co-aggregation of poly-GR/PR with other interactors identified in vitro. From 22 tested proteins, six showed convincing staining of endogenous proteins, but we could not detect co-localization with poly-GR in *C9orf72* patients (Fig S5A; see the Materials and Methods section).

Thus, among the tested poly-GR/PR–interacting proteins, STAU2 and the cytosolic ribosome seems to be the most relevant co-aggregating protein (complex) in *C9orf72* patients.

### Poly-PR reduces cytosolic ribosome levels and inhibits overall translation

Given the wide-spread interaction of poly-GR/PR with ribosomes and other RNA-binding proteins, we also analyzed global protein

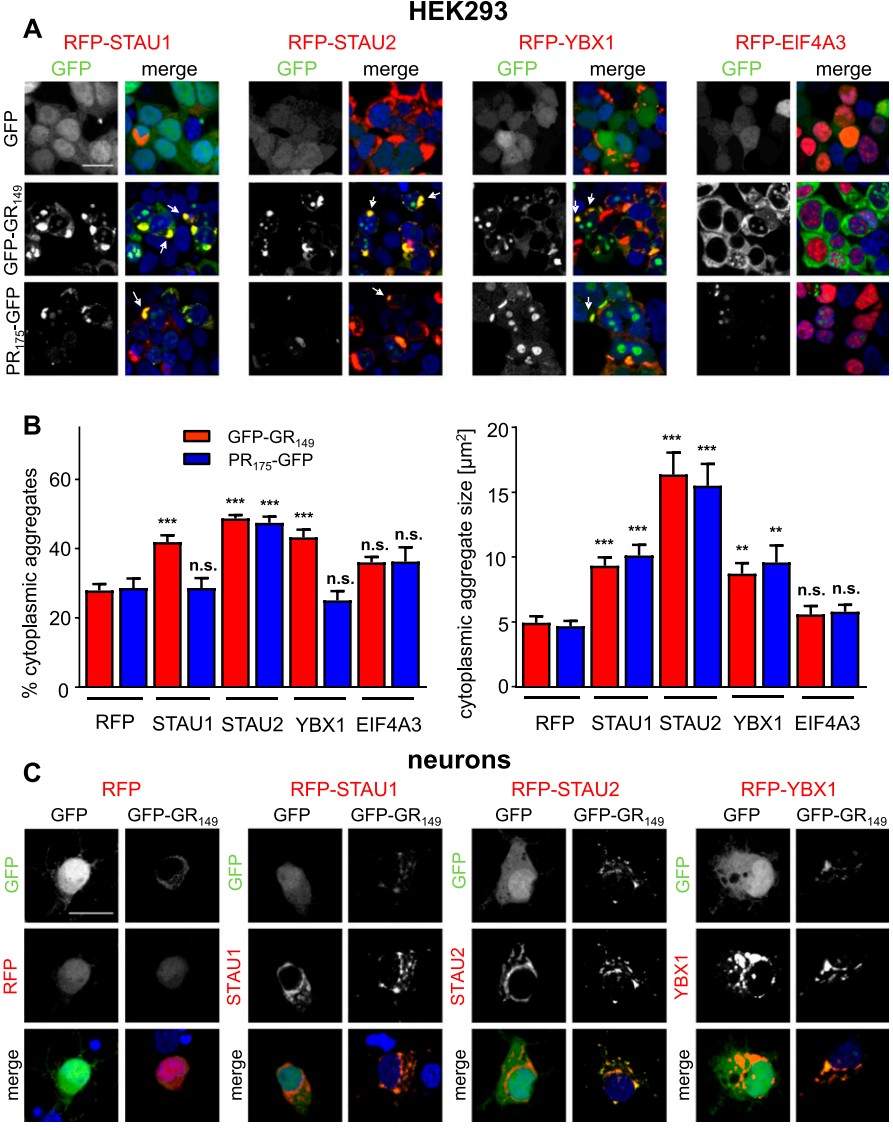

**HEK293**

**neurons**

**Figure 3. STAU1/2 and YBX1 recruit poly-GR/PR into large cytoplasmic granules.**
HEK293 cells and primary cortical neurons were co-transfected with GFP, GFP-(GR)$_{149}$, or (PR)$_{175}$-GFP expression vectors and RFP-tagged interactors associated with stress granules. **(A)** Immunofluorescence images of HEK293 cells showing co-expression of RFP-STAU1/2, RFP-YBX1, or RFP-EIF4A3 together with GFP-(GR)$_{149}$, (PR)$_{175}$-GFP, or GFP. STAU1/2 and YBX1 reroute poly-GR and poly-PR into large cytoplasmic structures (white arrows). Left columns show GFP signal, right columns show merge of GFP, RFP-tagged proteins, and nuclear DAPI staining (blue). Scale bar denotes 20 µm. **(B)** Quantifications of cytoplasmic poly-GR/PR granules from (A). Left bar graph shows percentage of cytoplasmic granules out of all granules (in nucleolus and cytoplasm) in poly-GR (red bars) and poly-PR (blue bars) (GR/RFP, n = 16 images (40×) from two independent experiments: PR/RFP, n = 14; GR/RFP-STAU1, n = 19; PR/RFP-STAU1, n = 20; GR/RFP-STAU2, n = 10; PR/RFP-STAU2, n = 8; GR/RFP-YBX1, n = 14; PR/RFP-YBX1, n = 14; GR/RFP-EIF4A3, n = 6; and PR/RFP-EIF4A3, n = 6). Cytoplasmic granule size is represented by the right bar graph (GR/RFP, n = 130 aggregates from two individual experiments; PR/RFP, n = 132; GR/RFP-STAU1, n = 104; PR/RFP-STAU1, n = 132; GR/RFP-STAU2, n = 123; PR/RFP-STAU2, n = 65; GR/RFP-YBX1, n = 119; PR/RFP-YBX1, n = 53; GR/RFP-EIF4A3, n = 93; and PR/RFP-EIF4A3, n = 71). Mean ± SEM is shown, exact $P$-values for left graph: GR/RFP-STAU1 versus GR/RFP, $P$ = 0.0001; GR/RFP-STAU2 versus GR/RFP, $P$ = 0.0001; GR/RFP-YBX1 versus GR/RFP, $P$ = 0.0001; GR/RFP-EIF4A3 versus GR/RFP, $P$ = 0.0775; PR/RFP-STAU1 versus PR/RFP, $P$ = 0.9999; PR/RFP-STAU2 versus PR/RFP, $P$ = 0.0007; PR/RFP-YBX1 versus PR/GFP, $P$ = 0.7820; PR/RFP-EIF4A3 versus PR/RFP, $P$ = 0.3936; exact $P$-values for right graph: GR/RFP-STAU1 versus GR/RFP, $P$ = 0.0001; GR/RFP-STAU2 versus GR/RFP, $P$ = 0.0001; GR/RFP-YBX1 versus GR/RFP, $P$ = 0.0023; GR/RFP-EIF4A3 versus GR/RFP, $P$ = 0.9672; PR/RFP-STAU1 versus PR/RFP, $P$ = 0.0001; PR/RFP-STAU2 versus PR/RFP, $P$ = 0.0001; PR/RFP-YBX1 versus PR/RFP, $P$ = 0.0023; and PR/RFP-EIF4A3 versus PR/RFP, $P$ = 0.8492 in one-way ANOVA with Dunnett's posttest. **(C)** Immunofluorescence images of transfected neurons (DIV7 + 3) co-expressing RFP-STAU1, RFP-STAU2, or RFP-YBX1, and GFP-(GR)$_{149}$ or GFP obtained by confocal microscopy are shown. Top row shows GFP signal, middle row shows RFP-tagged interactor or RFP control, and bottom row shows merge including nuclear DAPI (blue). Comparison of the largely homogenous poly-GR pattern in the RFP with punctate distribution in neurons co-expressing RFP-STAU1/2 and YBX1.

expression using quantitative LC-MS/MS in poly-GR/PR–transduced primary neurons. GFP-(GR)$_{149}$ transduction had no overt effect on the neuronal proteome and expression levels of none of its interactors were significantly altered (Fig S6A, red dots). Strikingly, (PR)$_{175}$-GFP expression significantly affected expression of hundreds of proteins compared with the GFP control. These changes were much larger than mRNA expression changes reported recently (Kramer et al, 2018), supporting a primary effect of poly-PR on translation. These findings are consistent with the selective toxicity of (PR)$_{175}$-GFP compared with GFP-(GR)$_{149}$ observed in our culture system (Fig S1C). GO annotation analysis revealed overall reduction of cytosolic ribosomal proteins, which may explain the strong down-regulation of synaptic and axonal proteins (Fig S6B). In contrast, the levels of nucleolar and mito-chondrial proteins were slightly increased. Overall, stress granule proteins (Fig S6B) and poly-GR/PR interactors (Fig S6A) were not affected. Despite the few large individual changes in the proteome of

poly-GR–expressing cells, enrichment analysis shows a small but significant overall reduction of ribosomal proteins (Fig S6B), which is consistent with the interaction of poly-GR with ribosomal proteins in cellular models and patient brains (Figs 1 and 5).

To substantiate this finding, we analyzed ribosomal protein expression in poly-PR/GR–expressing neurons by immunoblotting. Lentiviral poly-PR expression in primary neurons significantly reduced expression of ribosomal subunits RPS6, RPL19, and RPL36A, whereas cytoplasmic poly-GR expression had no strong effect (Fig 6A and B), despite the finding of subtle reduction by proteomics (Fig S6B). To investigate whether this loss of ribosomal components had an effect on total protein synthesis, we performed a surface sensing of translation (SUnSET) assay, which measures puromycin incorporation into newly synthesized proteins. After a brief pulse with puromycin, robust puromycin incorporation could be detected by immunoblotting with a puromycin-specific antibody. Importantly,

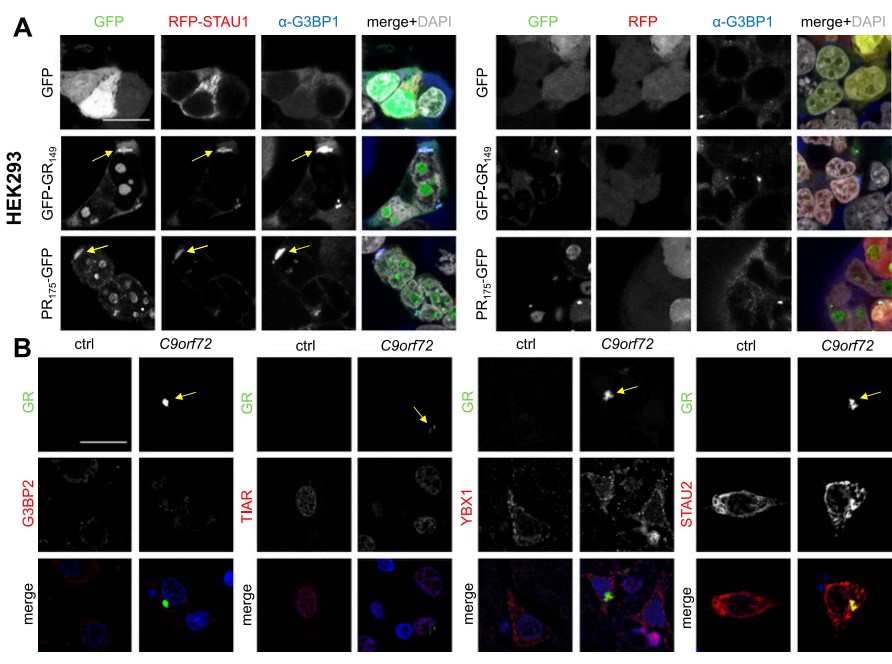

**Figure 4.  Cytoplasmic poly-GR/PR inclusions resemble stress granules in vitro.**
Immunofluorescence of stress granule markers in HEK293 cells and patient brain. DAPI visualizes nuclei. Single confocal planes were taken. Scale bar depicts 20 μm. **(A)** Co-localization of poly-GR/PR with the stress granule marker G3BP1 in HEK293 cells co-transfected with RFP-STAU1 or RFP control and DPR-GFP or GFP control. Left three columns show individual channels as indicated. The fourth columns show merge with additional nuclear DAPI staining in white. Arrows indicate cytoplasmic inclusions co-labeled with G3BP1. **(B)** Immunofluorescence of frontal cortex of a *C9orf72* patient and a healthy control case to analyze co-localization of poly-GR with stress granule components TIAR, G3BP2, YBX1, and STAU2. Arrows indicate poly-GR aggregates.
Source data are available for this figure.

the poly-PR–induced loss of ribosomal subunits was accompanied by a comparable reduction in overall protein synthesis compared with the GFP control, suggesting that the reduction of individual ribosomal subunits reflects a loss of functional ribosomes (Fig 6C and D).

Thus, poly-PR and to a lesser extent GFP-(GR)$_{149}$ expression leads to an overall reduction of cytosolic ribosomes, which results in a significant reduction of overall translation on poly-PR expression.

### Nucleolar poly-GR expression impairs translation and nucleolar structure and enhances toxicity

Numerous groups have reported poly-GR toxicity in various model systems, including primary neurons (Wen et al, 2014); these reports mostly used constructs with 20–100 repeats and typically involved predominantly nucleolar poly-GR localization, like we had observed in HEK293 cells, where we noticed slowed growth comparable with (PR)$_{175}$-GFP–expressing cells (Fig S1). Thus, we asked whether lack of nucleolar localization for GFP-(GR)$_{149}$ in primary neurons despite a significant overlap of interacting proteins with (PR)$_{175}$-GFP might explain these discrepancies. Therefore, we truncated our GFP-(GR)$_{149}$ construct resulting in GFP-(GR)$_{53}$ and then analyzed its localization in primary neurons. GFP-(GR)$_{53}$ showed diffuse cytoplasmic localization and strong localization in the nucleolus of 77.5% of the transduced neurons (Fig 7A). GFP-(GR)$_{53}$ also induced neuronal death compared with GFP control, although less effectively than (PR)$_{175}$-GFP (Fig 7B).

Moreover, lentiviral GFP-(GR)$_{53}$ expression also significantly reduced the expression of the ribosomal subunit RPS6 and protein synthesis similar to (PR)$_{175}$-GFP, whereas GFP-(GR)$_{149}$ had no effect, suggesting that nucleolar poly-GR/PR expression interferes with ribosomal biogenesis resulting in impaired translation and poly-GR/PR in vitro toxicity (Fig 7C and E). Because acute GFP-(GR)$_{53}$

toxicity is still weaker than poly-PR toxicity, we additionally analyzed nucleolus organization using immunofluorescence of fibrillarin (Fig 7A and D). In GFP-(GR)$_{149}$– and GFP–expressing neurons, most nucleoli showed homogenous staining of fibrillarin. In contrast, GFP-(GR)$_{53}$ expression led to a ring-like fibrillarin distribution and occasionally to a granular pattern, which was even more pronounced in (PR)$_{175}$-GFP–expressing neurons. Thus, nucleolar localization may promote the acute toxicity of poly-GR/PR seen in vitro.

## Discussion

We analyzed poly-GR/PR interactors in primary neurons and C9orf72 brains to address the disconnect between acute toxicity in various model systems and prodromal expression decades before clinical symptoms in patients. In primary neurons, poly-GR and poly-PR interact with RNA-binding proteins, including many components of the nucleolus, stress granules, and the ribosome. Overexpression of the interactors NPM1 and STAU1/2 reroutes poly-GR into the nucleolus or large stress granule–like structures in vitro, respectively. Poly-GR/PR toxicity in vitro depends on nucleolar localization and structural alterations of the nucleolus. Direct binding of ribosomes and/or impaired ribosomal biogenesis in the nucleolus likely contributes to impaired translation. Importantly, we detected ribosomal proteins in the pathognomonic neuronal cytosolic poly-GR/PR inclusions in *C9orf72* patients, suggesting that milder effects on translation may drive the slower neurodegeneration seen in patients.

### Poly-GR/PR interactome

We analyzed the interactome of poly-GR/PR in transduced rat cortical neurons and transfected HEK293 cells using GFP-(GR)$_{149}$

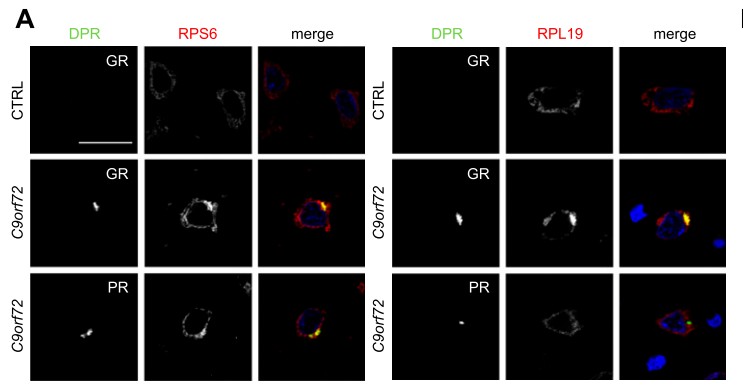

**Figure 5. Poly-GR and poly-PR co-aggregate with ribosomal proteins in *C9orf72* patients.**
**(A)** Immunofluorescent stainings of components of the small (RPS6) and large (RPL19) ribosomal subunits in *C9orf72* patient brains and controls. Additional ribosomal proteins are shown in Fig S4B. Note the enrichment of ribosomal proteins in poly-GR/PR inclusions. DAPI marks nuclei in blue. Single confocal planes were taken. Scale bar depicts 20 μm. **(B)** Quantitative analysis of co-localization of ribosomal proteins with poly-GR aggregates (n = 3 sections with 100 poly-GR aggregates counted each from *C9orf72* cortex, mean ± SEM is shown).

and $(PR)_{175}$-GFP baits. The repeat length of our constructs is still shorter than the repeats seen in patients but significantly longer than in previous studies using mainly pull-down with 20-mer or 30-mer peptides (Kanekura et al, 2016; Lin et al, 2016; Boeynaems et al, 2017; Yin et al, 2017) or recombinant expression of $(GR)_{50}$, $(GR)_{80}$, and $(PR)_{50}$ in cell lines (Lee et al, 2016; Lopez-Gonzalez et al, 2016). Moreover, the predominant cytoplasmic localization of GFP-$(GR)_{149}$ more accurately reflects the patient situation. Similar to the peptide-based studies, we identified a large number of RNA-binding proteins, in particular components of the cytosolic and mitochondrial ribosome, stress granules, the nucleolus, and (especially in poly-PR) splicing factors. A large fraction of proteins contains low-complexity domains associated with phase separation properties (Lee et al, 2016; Lin et al, 2016; Boeynaems et al, 2017).

It is striking that GFP-$(GR)_{149}$ and $(PR)_{175}$-GFP constructs interacted mostly with overlapping proteins but showed dramatically different toxicity in LDH release assays and effects on proteome composition. Quantitative comparison of the poly-GR and poly-PR interactomes is confounded by the different localization of poly-GR in HEK293 cells and primary neurons. Nevertheless, the methylosome (PRMT5/ WDR77), a component of the signal recognition particle (SRP72), and a stress granule marker (G3BP2) are specifically associated only with poly-GR in both cell types according to the most stringent criteria. GO analysis of pathways (Tyanova et al, 2016) enriched in both poly-GR interactomes further highlights the role of cytosolic translation in poly-GR toxicity. In patients, differential analysis of poly-GR/PR toxicity is difficult because poly-PR almost completely co-aggregates with poly-GR (Mori et al, 2013a). Although poly-PR interactors in neurons and HEK293 cells were distinct, GO analysis shows a clear enrichment of splicing factors in both cell types, which is consistent with reported effects on splicing (Kwon et al, 2014). In primary neurons, enrichment analysis for poly-PR is additionally confounded by strong down-regulation of many proteins including ribosomal subunits. Interestingly, yeast screens also identified vastly different modifiers for poly-GR and poly-PR (Jovicic et al, 2015; Chai & Gitler, 2018). Unexpectedly, knockout of several nonessential ribosomal subunits rescued poly-GR toxicity in yeast, but whether this may be primarily caused by reduced poly-GR expression was not addressed.

### Poly-GR/PR interactions with stress granules and the nucleolus

None of the previous interactome studies has reported co-aggregation of binding partners in poly-GR/PR inclusions in patients but focused their validation efforts on the effect of poly-GR/PR on the dynamics of membrane-less organelles, such as stress granules and the nucleolus (Lee et al, 2016; Boeynaems et al, 2017). Here, we addressed how the interactors affect poly-GR/PR, as phase separation could lead to aggregation of poly-GR/PR or the interacting proteins (Shin & Brangwynne, 2017).

Previously, the interactors STAU1/2 and YBX1 have been found in stress granules by co-localization analysis and proteomics (Thomas et al, 2009; Somasekharan et al, 2015; Jain et al, 2016). The dsRNA-binding proteins STAU1/2 are key components of RNA transport granules and help to dissolve stress granules in the recovery phase (Thomas et al, 2009), which may be impaired by binding to poly-GR/ PR. In addition, YBX1 was found to promote stress granule formation indirectly through induction of G3BP1 translation (Somasekharan et al, 2015). Here, we report that co-expression of STAU1/2 and YBX1 leads to formation of large cytoplasmic stress granule–like poly-GR/PR clusters. Whether the DPR proteins are recruiting stress granule proteins or the other way around is not clear. The poly-PR interactor EIF4A3 was also found in stress granules but had no effect on poly-GR/PR localization. Surprisingly, we could detect only STAU2 but not YBX1 or classical stress granule markers (G3BP1 and TIAR) in the poly-GR aggregates in *C9orf72* patients, suggesting that STAU2 binding may contribute to aggregation of poly-GR/PR aggregates in vivo. Moreover, interaction of poly-GR/PR with stress granule proteins may affect translation indirectly (Lee et al, 2016). We cannot exclude that other RNA-binding proteins interacting with poly-GR/PR contribute to the reduced translation.

Poly-GR/PR interact with several key nucleolar proteins, including NPM1 and NOP56. Overexpression of NPM1 recruited the predominantly cytosolic GFP-$(GR)_{149}$ into the nucleolus in primary neurons, whereas NOP56 had no such effect. Importantly, NPM1 was shown to induce phase separation of $(GR)_{20}$ and $(PR)_{20}$ in vitro (Lee et al, 2016). Super-resolution microscopy shows that poly-GR/ PR specifically localize to the NPM1-positive liquid-like granular component of the nucleolus (Lee et al, 2016). Our finding that nucleolar poly-$(GR)_{53}$ and especially poly-PR alter the distribution of fibrillarin and NOP56 within the nucleolus suggests that nucleolar poly-GR/PR may interfere with ribosomal biogenesis in vitro, which depends on NOP56 (Gautier et al, 1997). Importantly, nucleolar localization has not been detected for any DPR species in patients (Schludi et al, 2015; Vatsavayai et al, 2016) and the longer GFP-$(GR)_{149}$ localizing predominantly to the cytoplasm was not acutely toxic. However, poly-GR–bearing neurons in patients have

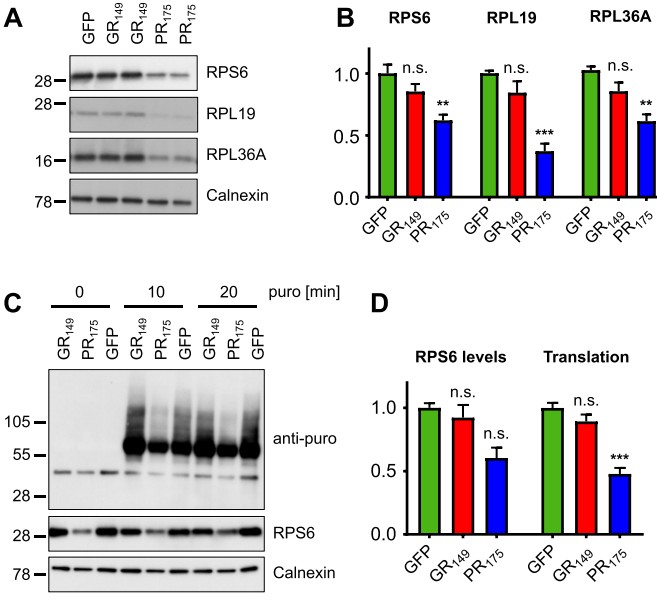

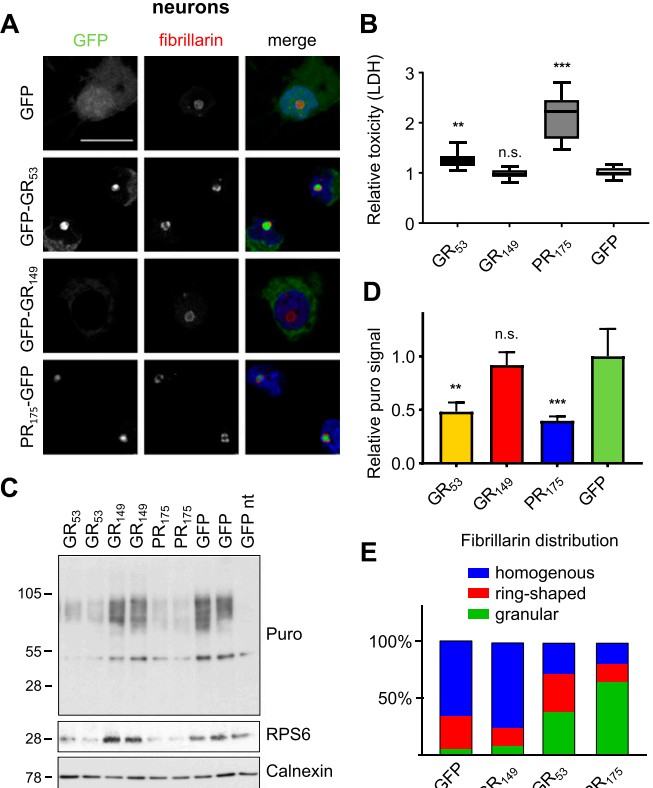

**Figure 6. Poly-PR inhibits translation in neurons.**
Primary rat cortical neurons (DIV6 + 7) were transduced with GFP, GFP-(GR)$_{149}$, or (PR)$_{175}$-GFP lentivirus. **(A)** Immunoblots show reduced expression of several ribosomal proteins in (PR)$_{175}$-GFP–expressing neurons. Calnexin was used as loading control. **(B)** Quantification of RPS6 signal normalized to calnexin (n = 6 from three independent experiments, mean ± SEM, exact $P$-values: GFP versus GFP-GR$_{149}$, $P$ = 0.1101 and GFP versus PR$_{175}$-GFP, $P$ = 0.0010 in one-way ANOVA with Dunnett's posttest), RPL19 signal normalized to calnexin (n = 6 from three independent experiments, mean ± SEM, exact $P$-values: GFP versus GFP-GR$_{149}$, $P$ = 0.1863 and GFP versus PR$_{175}$-GFP, $P$ = 0.0001 in the Kruskal–Wallis test with Dunn's posttest), and RPL36A signal normalized to calnexin (n = 6 from three independent experiments, mean ± SEM, exact $P$-values: GFP versus GFP-GR$_{149}$, $P$ = 0.1487 and GFP versus PR$_{175}$-GFP, $P$ = 0.0013 in the Kruskal–Wallis test with Dunn's posttest). **(C)** To quantify global translation, primary neurons were incubated with 1 μM puromycin (puro) for 0, 10, and 20 min before sample preparation, which is incorporated into truncated proteins (SUnSET system). A puromycin-specific antibody shows reduced levels of newly synthesized proteins in poly-PR–expressing neurons. Immunoblot for RPS6 and calnexin used as loading control. **(D)** Quantification of RPS6 signal normalized to calnexin (n = 3, mean ± SEM, Kruskal–Wallis test with Dunn's posttest, exact $P$-values: GFP versus GFP-GR$_{149}$, $P$ = 0.9999 and GFP versus PR$_{175}$-GFP, $P$ = 0.0507) and puromycin signal normalized to calnexin (n = 6, mean ± SEM, one-way ANOVA with Dunnett's posttest, exact $P$-values: GFP versus GFP-GR$_{149}$, $P$ = 0.2265 and GFP versus PR$_{175}$-GFP, $P$ = 0.0001).

larger nucleoli (Mizielinska et al, 2017), suggesting that more subtle nucleolar effects may be at play in patients. Investigating nucleolar organization in patient tissues may be rewarding.

Lee et al (2016) performed epistasis experiments with a large number of poly-GR/PR interactors originally identified in HEK293 cells using RNAi-mediated knockdown in flies. Interestingly, NPM1 knockdown reduced poly-GR toxicity in flies, whereas G3BP1 knockdown strongly enhanced poly-GR toxicity. Although they did not analyze poly-GR localization or aggregation under these conditions, the data are consistent with our hypothesis that poly-GR/PR in the nucleolus is most toxic by inhibiting ribosomal biogenesis, whereas sequestration of poly-GR/PR in stress granules may even be somewhat protective.

## Poly-GR/PR bind ribosomes and inhibit translation

We could detect the down-regulation of ribosomal proteins in poly-PR and to a lesser extent in GFP-(GR)$_{149}$–expressing neurons (Figs 6

**Figure 7. Nucleolar poly-GR/PR alter nucleolar organization and inhibit translation.**
GFP, GFP-(GR)$_{53}$, GFP-(GR)$_{149}$, or (PR)$_{175}$-GFP were transduced in primary rat neurons. **(A)** Images show fibrillarin immunofluorescence staining of hippocampal neurons. Left two columns represent GFP signal and fibrillarin staining in different DPR species as indicated. Right column shows merge with additional nuclear DAPI staining in blue. Scale bar denotes 20 μm. **(B)** LDH release assay detects significant cell death on lentiviral expression of (PR)$_{175}$-GFP and GFP-(GR)$_{53}$ but not GFP-(GR)$_{149}$ compared with GFP control in primary rat neurons (DIV7 + 14) (n = 3 independent experiments with six replicates each; box plot is shown with 25th percentile, median, and 75th percentile; and whiskers represent minimum and maximum; exact $P$-values: GFP versus GFP-GR$_{53}$, $P$ = 0.0011; GFP versus GFP-GR$_{149}$, $P$ = 0.9954; and GFP versus PR$_{175}$-GFP, $P$ = 0.0001 in one-way ANOVA with Dunnett's posttest). **(C)** SUnSET experiment in GFP-(GR)$_{53}$–, GFP-(GR)$_{149}$–, (PR)$_{175}$-GFP–, or GFP–expressing primary cortical neurons (DIV6+7) as in Fig 6C. Cells were incubated with 1 μM puromycin (puro) for 10 min or not treated (nt). Note the reduced incorporation of puromycin in neurons expressing GFP-(GR)$_{53}$ and (PR)$_{175}$-GFP. **(D)** Quantification of puromycin signal normalized to calnexin (n = 6 from three independent experiments, mean ± SEM, exact $P$-values: GFP versus GFP-GR$_{53}$, $P$ = 0.0022; GFP versus GFP-GR$_{149}$, $P$ = 0.8638; and GFP versus PR$_{175}$-GFP, $P$ = 0.0005 in one-way ANOVA with Dunnett's posttest). **(E)** Quantification of fibrillarin distribution within the nucleolus from (A). n = 6 to 16 images were analyzed.

and S6), leading to reduced translation efficiency (SUnSET assay) and an overall loss of structural proteins and cell death in poly-PR–expressing neurons. Strikingly, the shorter GFP-(GR)$_{53}$ variant, which localized to the nucleolus, was toxic and strongly reduced ribosome levels in neurons. The reduced levels of ribosomal proteins and the altered organization of the nucleolus suggest that poly-GR/PR toxicity in vitro is due to impaired ribosomal biogenesis. However, the neuronal interactome of cytoplasmic GFP-(GR)$_{149}$ strongly indicates that at least poly-GR also binds already-assembled

ribosomes. Two recent proteomics studies have reported predominant interaction of $(GR)_{20}$, $(GR)_{80}$, and $(PR)_{20}$ with cytosolic and mitochondrial ribosomes (Kanekura et al, 2016; Lopez-Gonzalez et al, 2016) but have not analyzed patient tissue. Here, we detected robust co-localization of several ribosomal proteins with ~30% of cytoplasmic inclusions in *C9orf72* patients, highlighting the physiological significance of the ribosomal pathway for *C9orf72* pathogenesis. The next important step will be to directly show reduced translation in a mouse model or patient tissue depending on poly-GR/PR inclusions.

In addition, direct binding of poly-GR/PR to mRNA may inhibit ribosomal access and thus translation (Kanekura et al, 2016), but stronger inhibition of translation by nucleolar GFP-$(GR)_{53}$ than cytoplasmic GFP-$(GR)_{149}$ argues against this hypothesis. A recent transcriptomics study reported slightly reduced expression of many ribosomal proteins in poly-PR–expressing neurons, consistent with reduced ribosomal biogenesis (Kramer et al, 2018). Because toxicity depends on nucleolar localization and reduction of ribosomal proteins, we propose that poly-GR/PR mainly interferes with ribosomal assembly/availability, but we cannot exclude additional effector mechanisms because translation is regulated at many levels. Also, the interaction of purified poly-GR/PR and ribosomes should be analyzed in greater detail using biophysical methods. It will be interesting to test whether restoring translation genetically or pharmacologically rescues poly-PR toxicity.

## Conclusion

Several hypotheses have been put forward to explain the toxicity of arginine-rich DPR proteins in various model systems, including impaired nucleocytoplasmic transport (Jovicic et al, 2015), oxidative stress (Lopez-Gonzalez et al, 2016), interference with membrane-less organelles (Lee et al, 2016; Lin et al, 2016; Boeynaems et al, 2017), impaired splicing (Kwon et al, 2014; Lin et al, 2016), and translation (Kanekura et al, 2016). From our findings, using poly-GR constructs of different length, it seems that altered nucleolar organization and impaired ribosomal biogenesis may be the most important mechanism of acute toxicity in vitro. Our discovery of several ribosomal proteins in cytoplasmic DPR inclusions in patient brains suggests that translation may be impaired by direct binding in vivo. Because poly-GR/PR inclusions are found many years before disease onset in a stage with moderate prodromal brain atrophy (Rohrer et al, 2015; Vatsavayai et al, 2016), the effects are most likely less acute than in most in vitro systems. Moreover, recruitment of poly-GR/PR into large stress granule–like structures by overexpression of some interactors with low-complexity domains and detection of STAU2 in patient aggregates suggests that phase separation may be a relevant driver of DPR aggregation. In particular, differential expression of poly-GR/PR interactors may, therefore, explain regional neurodegeneration despite widespread DPR expression. Overall, trapping of ribosomes in poly-GR/PR inclusions is the most direct link between poly-GR/PR in vitro toxicity and patient neuropathology and suggests a role of impaired translation in *C9orf72* pathogenesis.

# Materials and Methods

## DNA constructs and viral packaging

Synthetic genes with alternative codons for DPR sequences (GFP-$(GR)_{53}$, GFP-$(GR)_{149}$, and $(PR)_{175}$-GFP) containing an ATG start codon were subcloned into a pEF6/V5-His vector (Life Technologies) or a lentiviral vector driven by the human synapsin promoter (FhSynW2) (May et al, 2014). GFP from pEGFP-N1 (Clontech Laboratories) was used as negative control and subcloned into the respective vectors. Poly-GR/PR–interacting proteins were fused to the C-terminus of tagRFP by subcloning into FU3a-tagRFP. Empty vector was used as a control. Lentiviruses were packaged in human embryonic kidney cells (HEK293FT; Life Technologies) as described previously (Schwenk et al, 2014).

## Cell culture, transfection, and transduction

HEK293FT cells were cultivated in DMEM with Glutamax (Life Technologies) supplemented with 10% FCS, 1% Penicillin/Streptomycin, and 1% non-essential amino acids at 37°C with 5% $CO_2$. Cells were transfected with Lipofectamine 2000 (Invitrogen) following the manufacturer's instructions.

Primary cortical and hippocampal neurons were cultured from embryonic day 19 Sprague–Dawley rats and cultivated in Neurobasal Medium (Life Technologies) supplemented with 2% B27 (Life Technologies), 1% Pen/Strep, and 2 mM Glutamine. Primary neurons were transduced at 7 days in vitro (DIV7) using specified lentiviruses.

## SUnSET assay and immunoblotting

To analyze total protein synthesis, a SUnSET assay was performed. Therefore, primary cortical neurons were treated with 10 μg/ml puromycin (Merck) for 10 min at 37°C and 5% $CO_2$.

For immunoblotting experiments, neurons were harvested in 2× Laemmli buffer 7 days after transduction (DIV7 + 7). Samples were incubated at 95°C for 5 min and run on a 12% SDS–PAGE or 10%–20% tricine gels (Novex). The following primary antibodies were used for immunoblotting: anti-calnexin (ADI-SPA-860F; Enzo Life Technologies), anti-puromycin (clone 12D10, MABE343; Merck Millipore), anti-RPS6 (sc-74459; Santa Cruz Biotechnology), anti-RPL19 (sc-100830; Santa Cruz Biotechnology), and anti-RPL36A (sc-100831; Santa Cruz Biotechnology). For quantitative analysis, ImageJ was used and statistical analysis was done using the GraphPad Prism (version 7.01) software.

## Immunostaining and imaging in cell culture

For immunostaining, cells grown on PDL-coated glass coverslips were fixed with 4% paraformaldehyde for 15 min and permeabilized (0.2% Triton X-100 and 50 mM $NH_4Cl$ in PBS) for 5 min. After blocking (30 min, 2% fetal bovine serum, 2% serum albumin, and 0.2% fish gelatin in PBS), the coverslips were incubated in primary antibody solution at RT for 1 h and washed with PBS. Finally, the cells were incubated in Alexa-coupled secondary antibody solution and

treated with DAPI or TO-PRO-3 for staining of the nuclei. Antibodies and reagents used were anti-RPS6 (sc-74459; Santa Cruz Biotechnology), anti-G3BP1 (ab181150; Abcam), anti-fibrillarin (ab5821; Abcam), DAPI (Roche Applied Science), and TO-PRO-3 (Thermo Fisher Scientific). Single-plane images were obtained on a confocal laser scanning LSM710 microscope (Carl Zeiss) with a 63× or 40× immersion objective. Image editing and particle analysis was carried out using ImageJ software, and for statistical analysis, GraphPad Prism (version 7.01) software was used.

## Patient samples and immunofluorescence patient stainings

All patient materials were provided by the Neurobiobank Munich, Ludwig Maximilians University of Munich. Paraffin-embedded brain sections were deparaffinated and rehydrated with xylene and ethanol. To retrieve the antigen, slides were boiled 4× for 5 min in 100 mM citrate buffer of pH 6.0 using a microwave. After a brief rinse with deionized water, the sections were washed in PBS/0.05% Brij35, followed by blocking with 2% fetal calf serum in PBS for 5 min. The tissue was incubated overnight at 4°C in primary antibody solution. The following antibodies were used: STAU2 (ab60724; Abcam), anti-YBX1 (ab12148; Abcam), anti-FMRP (ab17722; Abcam), anti-G3BP2 (ab86135; Abcam), anti-TIAR (sc-136266; Santa Cruz Biotechnology), anti-RPS6 (sc-74459; Santa Cruz Biotechnology), anti-RPL19 (sc-100830; Santa Cruz Biotechnology), anti-RPS25 (HPA031801; Atlas Antibodies), anti-RPL36A (sc-100831; Santa Cruz Biotechnology), anti-GTPBP4 (ab184124; Abcam), anti-NOP56 (HPA049918; Atlas Antibodies), anti-PRMT1 (ab73246; Abcam), anti-WDR77 (HPA027271; Atlas Antibodies), anti-MAGOHB (ab186431; Abcam), anti-TRA2A (ab72625; Abcam), anti-GR clone 7H1 detecting predominantly non-methylated and asymmetrically dimethylated poly-GR (Schludi et al, 2015), anti-PR clone 32B3 raised against non-methylated poly-PR (Schludi et al, 2015), and rabbit polyclonal (Mori et al, 2013a). Afterward, sections were washed twice in PBS/0.05% Brij35 before incubation with Alexa-coupled secondary antibodies for 1 h at RT. Next, the sections were washed again, treated with DAPI for 15 min, and washed twice in PBS/0.05% Brij35 and twice in PBS only. The tissue was incubated in Sudan Black for 1 min at RT, rinsed in PBS, and mounted with Fluoromount Aqueous Mounting Medium (Merck).

Antibodies for poly-GR/PR that did not show convincing staining in brain sections were anti-CCDC40 (ab121727; Abcam), anti-PABPC4 (ab101492; Abcam), anti-MRPS9 (ab187906; Abcam), anti-MRPS11 (HPA050345; Atlas Antibodies), anti-MRPS23 (ab154533; Abcam), anti-MRPL12 (ab58334; Abcam), anti-PRMT5 (ab31751; Abcam), anti-CAPZA (ab166892; Abcam), anti-MOV10 (ab60132; Abcam), anti-ODZ3 (ab198923; Abcam), anti-SH3KBP (ab151574; Abcam), anti-TRA2B (ab66901; Abcam), anti-NDUFS3 (ab110246; Abcam), anti-SRP72 (PA5-56994; Thermo Fisher Scientific), anti-SNRPD3 (ab121129; Abcam), and anti-SNRPD2 (PA5-27547; Invitrogen).

## Viability of primary neurons and HEK293FT cells

Toxicity assays in transduced primary cortical neurons (DIV7 + 14) and HEK293FT cells were performed in 96-well plates using the LDH Cytotox Non-Radioactive Cytotoxicity Assay (Promega) following the manufacturer's instructions. To assess cell viability in HEK293FT cells,

an XTT (Roche) assay was used according to the manufacturer's protocols. Cells were cultivated in a 96-well plate. Absorption was measured after 24-h incubation time. Statistical analysis was performed using GraphPad Prism (version 7.01) software.

## Immunoprecipitation of poly-GR and poly-PR aggregates in neurons and HEK293FT

Quadruplicates of GFP-$(GR)_{149}$–, $(PR)_{175}$-GFP–, or GFP-infected cortical neurons (DIV7 + 8) and transfected HEK293FT cells were harvested in Benzonase Nuclease (Sigma) containing lysis buffer (2% Triton X-100, 750 mM NaCl, and 1 mM $KH_2PO_4$). Cell lysates were rotated for 45 min at 4°C. 10% was kept for whole proteome analysis, whereas the remaining samples were centrifuged at 1,000 g for 5 min at 4°C. With GFP antibody (clone N86/38; Neuromab), preincubated Protein G Dynabeads (Life Technologies) were added to the rest of the supernatant and incubated for 3 h at 4°C. After three washing steps (in 150 mM NaCl, 50 mM Tris, pH 7.5, and 5% Glycerol), they were used for further sample preparation for mass spectrometry analysis.

## LC-MS/MS

Peptides were separated on an EASY-nLC 1000 HPLC system (Thermo Fisher Scientific) via in-house packed columns (75-μm inner diameter, 30-cm length, and 1.9-μm C18 particles [Dr. Maisch GmbH]) in a gradient of buffer A (0.5% formic acid in $H_2O$) to buffer B (0.5% formic acid in $H_2O$ and 80% acetonitrile) at 300 nl/min flow rate. For IPs, we increased the content of buffer B from 2% to 30% 85 min before increasing the concentration of buffer B to 95% to wash the column. For complete proteomes, we increased the content of buffer B from 5% to 30% 155 min before increasing the concentration of buffer B to 95% to wash the column. The column temperature was set to 60°C. A Quadrupole Orbitrap mass spectrometer (Scheltema et al, 2014) (Q Exactive HF; Thermo Fisher Scientific) was directly coupled to the LC via nano-electrospray source. The Q Exactive HF was operated in a data-dependent mode. The survey scan range was set from 300 to 1,650 $m/z$, with a resolution of 60,000 at $m/z$ 200. Up to the 15 most abundant isotope patterns with a charge greater than or equal to two were subjected to collision-induced dissociation fragmentation at a normalized collision energy of 27, an isolation window of 1.4 Th, and a resolution of 15,000 at $m/z$ 200. Dynamic exclusion to minimize resequencing was set to 30 s (proteome) or 20 s (IP). Data were acquired using Xcalibur software (Thermo Fisher Scientific).

## MS data analysis and statistics

To process MS raw files, we employed the MaxQuant software (v 1.5.3.54 for HEK data and 1.5.4.3 for neuron data) (Cox & Mann, 2008) and Andromeda search engine (Cox et al, 2011), against the UniProtKB rat FASTA database (08/2015) and UniProtKB human FASTA database (08/2015), respectively, using default settings. Enzyme specificity was set to trypsin, allowing cleavage N-terminally to proline and up to two mis-cleavages. Carbamidomethylation was set as fixed modification, and acetylation (N terminus) and methionine oxidation were set as variable modifications. A false

discovery rate (FDR) cutoff of 1% was applied at the peptide and protein level. For rat neuron data, the FDR was independently calculated and applied for IP samples and complete proteomes by setting individual parameter groups in MaxQuant. "Match between runs," which allows the transfer of peptide identifications in the absence of sequencing after nonlinear retention time alignment, was enabled with a maximum retention time window of 0.7 min. Protein identification required at least one razor peptide. Data were filtered for common contaminants (n = 247). Peptides only identified by site modification were excluded from further analysis. Proteins were normalized with MaxLFQ label-free normalization (Cox et al, 2014). The mass spectrometry proteomics data have been deposited to the ProteomeXchange Consortium via the PRIDE partner repository with the dataset identifier PXD008691 (Vizcaino et al, 2016).

For bioinformatic analysis and visualization, we used the open PERSEUS (v 1.5.2.12, 1.5.3.4, 1.5.4.2, 1.5.5.5, and 1.5.8.7) environment (Tyanova et al, 2016), MaxQuant (neurons: 1.5.4.3 and HEK: 1.5.3.54), and the R framework (Team, R Development Core, 2008). Imputation of missing values was performed with a normal distribution (width = 0.3 and shift = 1.8). For pairwise comparison of proteomes and determination of significant differences in protein abundances, $t$ test statistics were applied with a permutation-based FDR of 5% and S0 of 1 (Tusher et al, 2001), requiring at least 66% valid values in at least one group per comparison. For the 1D and 2D annotation, we first matched GO data (GOMF name, GOCC name, GOCC slim name, GOBP slim name, and Kegg and UniProt keywords) to the protein identifiers (major ID) in Perseus. Moreover, the annotation for stress granule proteins identified by Jain et al (2016) for humans was assigned both for HEK and rat neuron data. Stress granule annotations were transferred from human to rat for genes with identical gene names in both species. The 1D and 2D annotation enrichment was performed on the Welch's $t$ test difference in the Perseus environment. FDR control was performed using the Benjamini–Hochberg correction separately within each annotation category, e.g., GOCC or GOBP. Accordingly, the FDR cutoff (q-value < 5%) relates to slightly different $P$-values in the different annotation categories, thereby leading to significant (black) and insignificant (gray) populations slightly overlapping in the $-\log_{10}$ ($P$-value) dimension in the 1D annotation plots (Fig S6B). The 2D annotation plots show annotation terms with q-values < 0.1. Both 1D and 2D annotation terms were filtered for terms comprising at least six proteins quantified by mass spectrometry.

We assessed the content of low complexity in the neuronal interactome based on IUPred-L (Dosztanyi et al, 2005). We queried all proteins detected in the neuronal interactome data (see filtering criteria), significant poly-PR interactors, and significant poly-GR interactors in D2P2 (Oates et al, 2013). Queries were based on MaxQuant reported UniProt identifier, using the first entry if multiple identifiers were reported in protein groups. For determining significant differences, we employed the Mann–Whitney–Wilcoxon test.

## Supplementary Information

## Acknowledgements

We thank Irina Pigur for excellent technical assistance. We thank Christine Hösl for providing access to the confocal microscope when we were in need. We thank Christian Behrends, Dorothee Dormann, Kathrine LaClair, Carina Lehmer, Bettina Schmid, and Martin Schludi for critical comments on the manuscript. This work was supported by the Hans und Ilse Breuer Foundation (D Edbauer), the Munich Cluster of Systems Neurology (D Edbauer), the NOMIS Foundation (D Edbauer), and the European Community's Health Seventh Framework Programme under grant agreement SyG-318987 (ToPAG) (M Mann) and 259867 (EUROMOTOR) (M Mann) and 617198 (DPR-MODELS) (D Edbauer).

### Author Contributions

H Hartmann: conceptualization, formal analysis, investigation, visualization, methodology, and writing—original draft.
D Hornburg: conceptualization, formal analysis, investigation, visualization, and writing—review and editing.
M Czuppa: formal analysis and investigation.
J Bader: formal analysis, investigation and visualization.
M Michaelsen: resources.
D Farny: resources.
T Arzberger: resources.
M Mann: supervision and funding acquisition.
F Meissner: conceptualization and supervision.
D Edbauer: conceptualization, resources, supervision, funding acquisition, investigation, writing—original draft, project administration, and writing—review and editing.

### Conflict of Interest Statement

The authors declare that they have no conflict of interest.

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
