## [Reviewer comments · Life Science Alliance]

Proteomics and C9orf72 neuropathology identify ribosomes as poly-GR/PR interactors driving toxicity

Hannelore Hartmann, Daniel Hornburg, Mareike Czuppa, Jakob Bader, Meike Michaelsen, Daniel Farny, Thomas Arzberger, Matthias Mann, Felix Meissner, Dieter Edbauer
DOI: 10.26508/lsa.201800070

Review timeline:	Submission Date:	20 April 2018
	Revision Received:	20 April 2018
	1 st Editorial Decision:	23 April 2018
	Accepted	28 April 2018

Report:

(Note: Letters and reports are not edited. The original formatting of letters and referee reports may not be reflected in this compilation.)

Please note that the manuscript was previously reviewed at another journal and the reports were taken into account in inviting a revision for publication at *Life Science Alliance* prior to submission to *Life Science Alliance*.

1st Revision – authors' response

20 April 2018

Referee #1:

In this manuscript, Hartmann, Hornburg and colleagues used quantitative mass spectrometry to identify poly-PR/GR interacting proteins in primary rat neurons and HEK293T cells. In agreement with several other studies using a similar approach, the authors identify ribosomes, stress granules, and the nucleolus as enriched interactors with poly-GR/PR. The authors then assessed the relevance of selected interactors in cell-based disease models and in patient tissue through co-localization analysis.

The present study distinguishes itself from those previously published by assessing DPR interactomes in primary neurons, performing post-hoc analysis of interactions in patient tissue, and looking at the impact of poly-GR length on localization and toxicity. However, as the central conclusions drawn from their analysis are similar to those previously reported, our main concern is that this manuscript, in its current form, does not substantially advance the field's understanding of how arginine-rich DPRs contribute to toxicity in disease.

It is true that several previous studies analyzed poly-GR/PR interacting proteins in vitro, but none of these studies confirmed interactors in patient cells or tissues leaving the relevance of these findings unclear. We performed an extensive analysis of interactors in C9orf72 patients. We now show more negative results (new Fig. S5) for interactors and provide a list of other tested antibodies with poor staining quality in patient tissue in the methods section. From these data, we conclude that the ribosome and STAU2 are the most relevant interactors in patient tissue.

Major Concerns

1. The thesis of the paper seems to be that nucleolar poly-GR/PR localization mediates toxicity in cell based systems. They hint that this might be from altered ribosomal biogenesis based on their finding that nucleolar localization correlates with altered global protein translation rates. Establishment of this thesis with greater rigor would represent a more substantial advance. However, their current data is insufficient to reach this conclusion, especially given the multiple other ways in which these proteins might impede translation (through stress granule activation, for example). Below are suggestions that might aid this process

- a. *Does overexpression of NPM1, but not NOP56, enhance poly-GR-related toxicity and suppress protein translation in neurons? If so, this would suggest these phenotypes are more nucleolar-mediated.*

These points are all valid suggestions. However, the experiments will take at least 3 months because all experiments have to be done in primary neuron using lentiviral manipulation. Consistent with our hypothesis NPM1 knockdown reduced poly-GR toxicity in flies (Lee et al, Cell 2016).

- b. *STAU1/2 and YBX1, but not eIF4A3, lead to redistribution of Poly GR to cytoplasmic granules. Does this redistribution rescue the impaired growth phenotype or alter protein translation rates in poly-GR/PR expressing HEK293T cells or neurons? If so, this would suggest that driving localization away from the nucleolus would be relevant even at the cost of stress granule formation.*

Only transfection but not lentiviral transduction of STAU1/2 and YBX1 results in sufficient expression levels to re-localize poly-GR/PR into cytoplasmic granules in primary neurons (Fig. 3C). However, in the transfection-based approach we would have to use single-cell methods to analyze translation and toxicity because the transfection rate is only 1-2%.

- c. *Do STAU1/2 and YBX1 overexpression, but not eIF4A3, reduce nucleolar poly-PR concomitant with its increased incorporation in cytoplasmic granules, and, if so, does this reduce poly-PR-mediated neuronal toxicity?*

While we need high-level STAU1/2 and YBX1 expression to achieve poly-GR re-localization which is only possible with transfection, poly-PR transfection is apparently too toxic to detect any surviving cells. Therefore, the suggested experiment is unfortunately not feasible in neurons.

2. The authors argue that because ribosomal proteins co-localize with cytoplasmic GR and PR in patient tissue, the ribosomal pathway is important in disease pathogenesis. However, as the author's point out, PR and GR are not predominantly localized to the nucleolus in patient neurons, and in Figures 5 and 6, the authors' results strongly suggest that that nucleolar localization of these DPRs is necessary for reduced levels of ribosomal proteins and global translation inhibition in primary rat neurons. How do you reconcile these to seemingly conflicting observations?

We speculate that nuclear GR/PR interfere with ribosome assembly in cellular models, but the cytoplasmic GR/PR in patients interfere with ribosome function. Both could be mediated by direct binding to the (pre)ribosome. As indicated below, analyzing effects of poly-GR/PR on translation in patients is very difficult. However, our data is so far the most direct link between the various in vitro findings on poly-GR/PR toxicity and patient neuropathology.

3. Establishing that there is a functional ribosomopathy in C9 patient cells and tissues at endogenous expression levels would significantly strengthen the thesis of this paper.

- a. *Are there reduced levels of ribosomal proteins in cells or tissues derived from patients?*

- b. *Is there reduced translation in patient cells?*

- c. *Are there changes in nucleolar organization observed in patient cells?*

These are important questions that will take at least 6-9 months to answer because they require single cell analysis. In the immunofluorescence staining we did not detect overt reduction of cytoplasmic ribosomal proteins in aggregated-bearing cells (Fig. 5). In vitro translation experiments from patient tissue are impaired by the low frequency of poly-GR/PR inclusions in bulk tissue. Unfortunately, poly-GR/PR expression or even aggregate formation has not been robustly detected in patient fibroblasts or iPSC-derived neurons, which precludes use of patient cell lines. Mizielinska et al (Acta Neuropathol Commun, 2017) had reported increased nucleolar size in poly-GR bearing neurons in C9orf72 brains but had not addressed detailed organization of this organelle.

4. *Physical Interactions identified by immunoprecipitation highlighted for RNA binding proteins such as Staufen or specific Ribosomal proteins should be confirmed in the presence of RNase as both PR and GR are positively charged proteins which may bind specifically or non-specifically to nucleic acids. This would change our understanding of their interactions with these compartments and is thus important.*

This is a valid point. Kanekura et al. (HMG, 2016) reported indeed that GR/PR-peptides can directly bind to mRNA. RNase treatment studies would be difficult to interpret for the ribosome because RNase is also known to degrade the ribosomal RNA. Direct interaction can only be validated using purified ribosomes and poly-GR/PR using biophysical methods or cryo-electron tomography. These tedious studies are beyond the scope of a revision. We discussed this point in the revised manuscript. Finding only colocalization the ribosome and STAU2 in patient aggregates argues against unspecific RNA-mediated interaction of poly-GR/PR with the identified proteins.

Minor Concerns

5. *More direct comparison of the DPR interactomes in neurons and HEK293T cells would help establish how GR149's differential localization in the two cell types contributes to its differential effects on cell survival and growth, relative to PR175.*

Presenting and comparing hundreds of proteins graphically is very challenging. poly-GR/PR interactors identified in HEK293 and primary neurons are labeled in Fig 1 (italics). All identified proteins are provided in the supplemental tables for the reader's own study.

6. *The authors discuss select GO terms enriched within interactors, but do not provide a full list or enrichment scores.*

The enrichment scores for Fig 1B were already included in Table S2A/B. We additionally include the raw data for Fig. S6 in the new Table S3.

7. *In regards to Fig 4B, does RFP-eIF4A3 overexpression have no effect on GFP-GR149 distribution in neurons, as in HEK293T cells?*

We think this comment refers to Fig. 3C. As in HEK293T cells (Fig. 3A), eIF4A3 had no effect on poly-GR localization in neurons as shown below. This is now mentioned in the revised manuscript.

Non-essential suggestions

1. *Splitting Figure S3 into individual panels would make it easier for readers to locate data referenced in main text.*

We changed the figure as requested.

2. Clearer labeling of main figures would help readers know when experiments are performed in HEK293T cells vs. neurons.

We added this important information to all figures.

Referee #2:

GGGGCC repeat expansion is the most common genetic cause of both ALS and FTD. Since the translation products of G4C2 sense and antisense RNAs, such as poly-GA, poly-GR, and poly-PR, were found to form abnormal protein aggregates in patient brains in 2013, their toxicity has been under intense investigation by many labs using biochemical, molecular and genetic approaches in different cellular and animal models. In this descriptive study, interactome analysis in rat primary cortical neurons and HEK293 cells showed that poly-GR and poly-PR preferentially bind to many RNA binding proteins, including components of stress granule proteins, ribosomes, and nucleoli. Many studies have already reported very similar findings (Lee et al., Cell 2016; Lin et al., Cell 2016; Lopez-Gonzalez et al., Neuron 2016; Boeynaems et al., Mol. Cell 2017; Yin et al., Cell Reports 2017). Similarly, the finding on SG markers in Figure 4 has been reported previously (e.g., Lee et al., Cell 2016; Boeynaems et al., Mol. Cell 2017). Another recent study highlighted the interaction between poly-GR/poly-PR and ribosomes and inhibition of global translation (Kanekura et al., Hum. Mol. Genet. 2016). Thus, the finding that poly-GR/poly-PR inhibits translation (Figure 5) is not new neither. Overall, despite the authors emphasized multiple times about the minor differences between their approach and earlier published studies, such as the cell types and length of poly-GR/poly-PR proteins used, the current descriptive work provides limited conceptual advancement.

It is true that several previous studies analyzed poly-GR/PR interacting proteins in vitro, but none of these studies confirmed interactors in patient cells or tissues leaving the relevance of these findings unclear. We performed an extensive analysis of interactors in C9orf72 patients. We now show more negative results (new Fig. S5) for interactors and provide a list of other tested antibodies with poor staining quality in patient tissue in the methods section. From these data, we conclude that the ribosome and STAU2 are the most relevant poly-GR/PR interactors in patient tissue.

Another major concern is the artificial nature of some experimental observations. As the authors correctly point out, in most published studies, overexpressed poly-GR and especially poly-PR are predominantly located in nucleoli, "which is not observed in patient tissue". Knowing this, the authors went ahead anyway to use lentivirus to overexpress (PR)175-GFP and GFP-(GR)53, which localized to nucleoli. Both (PR)175-GFP and GFP-(GR)53 reduced ribosome levels and translation, consistent with the results of Kwon et al. (Science 2014) who first reported this toxic mechanism with 20-mers. Because the nucleolar localization "is not observed in patient tissue", this set of experiments may help explain the artificial acute toxicity observed in vitro but the relevance to human disease is questionable.

This is an important point that has been largely ignored in the previous studies. We speculate that nucleolar GR/PR interfere with ribosome assembly in cellular models, but the cytoplasmic poly-GR/PR in patients interferes with ribosome function as evidenced by co-aggregation of ribosomes in patient brain. Both effects could be mediated by direct interaction of poly-GR/PR with the (pre)ribosome. We have addressed this point clearer throughout the revised manuscript

The experiments to co-overexpress GR/PR-interacting proteins are also quite artificial. The effect of overexpression of STAU (and other proteins) itself on cytoplasmic granule formation should be examined first. Systematic analysis of GR/PR-interacting proteins and their potential colocalization with GR/PR aggregates in patient tissues is probably much more meaningful.

STAU1, STAU2 and YBX1 alone only show very little granule formation (also see G3BP1 staining for RFP-STAU1 expression, Fig. 4A first row) in control conditions. Examples for the other constructs are shown below. As indicated above, we have analyzed many poly-GR/PR interactors in patient tissue but could only confirm the ribosome and STAU2. We now provide a full list of tested antibodies in the method section and show more negative results in Fig. S5.

Other minor issues

1. The authors first used lentivirus to acutely overexpress GFP-(GR)149 and (PR)175-GFP in HEK293 cells and found they are localized in the nucleolus. As the authors pointed out, this may well be an experimental artifact. Did they try anything to improve the experimental system, such as using different systems to express these abnormal proteins at a lower level or with different tags? Since the relative expression levels of poly-GR in HEK293 cells versus neurons were not measured, one cannot conclude it is not toxic in neurons.

Expression levels are an important point that have been ignored in previous studies. Lentiviral expression in neurons likely results in lower protein levels than transfection in HEK293 cells. Transfection of neurons as in Fig. 3C certainly leads to higher expression levels than lentiviral expression but does not alter poly-GR expression, while increasing poly-PR toxicity.

2. It looks like poly-GR recruits STAU into cytoplasmic granules, but not the other way around as the authors concluded. Does STAU overexpression by itself cause some aggregate formation?

Indeed, STAU1 alone only shows very little granule formation (see G3BP1 staining in Fig. 4) in control GFP transfected HEK cells. However, STAU1/2 expression increase granule number (Fig. 3B).

Based on the stress granule localization of STAU1/2 we assume that STAU1/2 is driving the cytoplasmic clustering of poly-GR but we cannot formally exclude that it is the other way round but since the interaction of both proteins is required for the effect this a rather theoretical point. We clarified this in the text.

3. In Figure 3B, STAU1 and YBX1 have no effect on the number of cytoplasmic granules induced by PR. What does this negative result mean?

We cannot provide any rationale for this observation other than that only STAU2 is found in poly-GR/PR aggregates in patient tissue.

Referee #3:

In this paper, Hartmann et al. investigate whether the poly-GR and poly-PR proteins present in C9orf72 ALS/FTD are toxic in neurons, and generate some evidence supporting the hypothesis that these DPRs cause toxicity by reducing translation. This paper provides the first neuron-specific analysis of the interactomes of poly-GR and poly-PR, showing that the poly-GR and poly-PR in neurons bind many of the same low complexity domain-containing RNA binding proteins as they bind in HEK293T cells. The paper furthermore analyzes the effect of DPR expression on the proteome, showing that poly-PR expression alters the abundance of many proteins. The paper also addresses the question of why poly-GR and poly-PR are abundant in nucleoli in various model systems but not in the nucleoli of patients. It shows that while poly-GR localizes to the nucleolus in HEK293FT cells, it does not in primary neurons. Furthermore, it shows that a short poly-GR protein (similar to that used in many in vitro experiments) localizes to the nucleolus while a longer poly-GR protein (more comparable to the poly-GR present in patients) is predominantly cytoplasmic. Additional experiments investigating the subcellular localization of poly-GR/PR show that the overexpression of some poly-GR/PR interacting proteins can alter the localization of poly-GR/PR. Hartmann et al. hypothesize that when poly-PR/GR localize to the nucleolus in vitro, they diminish levels of ribosomal proteins and thus reduce translation. They also observe ribosomal subunits in DPR aggregates in patients, and conclude that translational repression might also occur in patients.

The authors argue that localization of DPRs to the nucleolus causes deficient ribosome biogenesis, which reduces translation rates, which in turn leads to cell death. While each of these claims is plausible, the authors do not convincingly support any of them. Nevertheless, this paper makes valuable observations. The observation that ribosomal proteins are enriched in DPR aggregates in patients is particularly intriguing, as is the finding that the length of DPRs used in in vitro experiments may be a major determinant of the behavior and subcellular localization of the DPR. The authors also do an excellent job of fitting their work into the broader picture by mentioning relevant findings in other studies.

Major points

- *Certain conclusions seem to be overstated.*

We worked on the text and now draw less strong conclusions.

o The authors conclude that the reduction in the levels of certain ribosomal proteins is the cause of reduced translation in the presence of poly-PR. This is an appealing hypothesis, but given that translation can be regulated by many other factors in addition to the availability of ribosomes, further experiments must be done before this conclusion can be drawn.

We had already mentioned that interaction of poly-GR/PR with stress granule proteins may affect translation indirectly but extended this point in the revised discussion.

o The link between nucleolar localization of DPRs and reduced ribosomal protein or protein synthesis is also overstated.

We toned down our conclusions in the result section.

o The link between reduced translation and cell death is also overstated.

We agree. Since acute GFP-(GR)₅₃ toxicity is still weaker than poly-PR toxicity, we think that nucleolar misorganization contributes to the acute poly-PR toxicity seen in vitro (Fig 7B/E). This is properly discussed in the revised manuscript.

o Evidence for reduced translation in patients is lacking.

We mention this limitation in the revised discussion.

• *The handling of statistics appears to be problematic:*

o The description of the number of images as an "n" (for example, in Figure 3B) is a bit misleading, as presumably each independent experiment is only represented by one average value in the statistical analysis. Technically, to comply with EMBO's requirement that "if n<5, the individual data points from each experiment should be plotted and any statistical test employed should be justified," Figure 3B should be replotted with the individual data points shown.

In all our data n represents the number of values per group used for the actual statistics. We analyzed in total n=6 to 20 pictures from two independent experiments. We did not first average the data from each experiment to do statistics on "n=2". Although n>5, graphs with individual data points are shown below, but do not provide more insight.

The fraction of cytoplasmic aggregates was quantified per image (n=6-20 images per group from 2 independent experiments). Raw data are shown below.

The statistics on aggregate size was initially done on n=53-132 individual aggregates (from 2 independent experiments). We now changed this to per-image analysis to be more consistent within the figure (n=5-17 images from 2 independent experiments). The different analysis did not affected the overall result and conclusion, but slightly changed the significance levels.

o While the decision to avoid parametric statistics when normality cannot be affirmed is laudable, it is not helpful to perform a Kruskal-Wallis test with an n of 3 (as in Figure 5D), because it is impossible to get a p-value of less than 0.05 with this test if n = 3.

Since the data was not distributed normally, we had to use the Kruskal-Wallis test. This test yields a highly significant difference for translation that is consistent with the robust effect (compare time series in panel 6C). The apparent reduction of RPS6 did not reach significance, but we show similar effect size with significant differences for other ribosomal subunits (e.g. Fig. 6B). The Kruskal-Wallis test is computationally very difficult for n>30 (https://en.wikipedia.org/wiki/Kruskal%E2%80%93Wallis_one-way_analysis_of_variance), but most natural data with high n should approach normal distribution according to the central limit theorem, which would allow parametric tests.

• Additional quantifications of the following points would be helpful:

o Could you provide quantification to support the claim that "the less frequent cytoplasmic poly-GR/PR punctae were predominantly G3BP1 positive"?

We did not do quantification of this but can show a representative image showing that cytoplasmic DPR aggregates are G3BP1 positive.

o Quantification of colocalization of G3BP2, TIAR, YBX1, and STAU2 with poly-GR inclusions should be performed.

We now provide this quantification in the text:

“We detected not a single poly-GR/PR inclusion convincingly colocalizing with classical stress granule markers proteins (G3BP2, TIAR) and the interactor YBX1 in two C9orf72 patients. However, ~25 % of poly-GR inclusions (76 of 300 counted aggregates) were co-stained with STAU2 in cortex (Fig. 4B).”

o Levels of RPS6, RPS2, RPL19, and RPL36A in transduced cortical neurons (Figure 5B) should be quantified.

The graph in Fig. 6B now provides the quantification.

o Quantify the enrichment of RPS6, RPS25, RPL19, and RPL36A in poly-GR and poly-PR inclusions in patient brain.

The graphs in Fig. 5B and S4B now contain the quantification.

o Puromycin incorporation in transduced neurons should be quantified.

Puromycin signal is now quantified in Fig. 7D. For Fig. 6D the quantification was already included in the original manuscript.

• *It would be valuable to expand upon the significance of certain findings:*

o What does it signify that STAU1/2 and YBX1 reroute DPRs while EIF4A3 does not?

o What is the significance of the altered distribution of fibrillarin in GFP-(GR)53 and (PR)175-GFP-expressing neurons?

o What might it mean that ribosomal subunits coaggregate with DPRs but classic stress granule markers do not?

Our experiments provide a basic validation of several poly-GR/PR interactors. Further follow-ups are certainly interesting, but beyond the scope of revision experiments. The finding of ribosomes and STAU2 but not stress granule markers in patient inclusions indicates to us that this interaction is most important for actual C9orf72 pathogenesis.

Minor points

• *To illustrate the downregulation of ribosomal proteins due to expression of poly-PR, it might be helpful to have a volcano plot similar to those in S5A, but specifically showing ribosomal proteins.*

We feel that the global annotation analysis provided in Fig. S6B is statistically more robust. The raw data for Fig S6A is available in Table S1 for the readers to investigate their favorite proteins.

• *Based on the image in S1A, it is not accurate to write (in the Results section), "In HEK293 cells both proteins predominantly localized to the nucleolus."*

We modified our conclusion in the text accordingly.

• *While ANOVA is an appropriate means by which to detect differences between groups, it is not an appropriate way to identify the absence of a difference. Therefore, strong statements ("In contrast, stress granule-associated poly-PR interactor EIF4A3 had no effect on poly-GR/PR localization or clustering") should be replaced with statements conveying that no difference was detected.*

We modified the text accordingly.

- *In the sentence, "These findings are consistent with the selective toxicity of...", Figure S1C, not Figure S1B, should be mentioned.*

We modified the text accordingly.

- *In the second panel in Figure 4B, the image does not include a GR aggregate, which makes it impossible to assess whether such aggregates are also TIAR positive.*

There is a small poly-GR aggregate visible. We labeled aggregates with arrows for clarity.

- *A more thorough description of what previous papers have revealed about the localization of poly-GR and poly-PR in C9orf72 patient brain would be helpful.*

We cite several previous studies regarding distribution in patient brain, such as Schludi et al. (Acta Neuropathologica, 2015) and Vatsavayai et al (Brain 2016) and Saberi et al (Acta Neuropathologica, 2018).

- *Analysis of the transcriptome of neurons transduced with poly-PR might rule out the possibility that alterations in the proteome are due to changes in transcription or RNA stability rather than translation.*

Transcriptomic analysis was recently reported by Kramer et al (Nature Genetics 2018). However, the expression changes were much larger on the protein level than on the mRNA level arguing for a primary deficit in translation. We mention this in the revised manuscript.

- *Please justify the use of calnexin as a loading control.*

We picked calnexin for no particular reason. Calnexin has been used by others as a loading control, e.g. Spencer ML et al, J Biol Chem. 2004.

- *Given that overexpression of NPM1 in neurons drives increased nucleolar localization of poly-GR, comparing LDH release in neurons expressing GFP-GR149 alone with those expressing GFP-GR149 and NPM1 might provide further evidence in support of a causal relationship between nucleolar DPRs and toxicity.*

These experiments are interesting but beyond the scope of a revision.

Non-essential comments/requests for clarification

- *In Figure 1A, do italics mean that the protein was identified in at least one out of the two HEK293 experiments (i.e., GR or PR)? Might there be a way to indicate whether a protein was found in the interactome of GFP-GR alone, PR-GFP alone, or both?*

The proteins in italics were also found in at least one of the HEK293 experiments. To keep the graph readable, we prefer not to indicate whether a protein was found in the interactome of GFP-GR alone and PR-GFP alone. All raw data are available in Table S1.

- *I don't quite understand the y-axis label in Figure 1B: is the graph showing the proportion of each protein that is of low complexity? Wouldn't that be "low complexity/length" rather than "length/low complexity"?*

We corrected the mistake. The plot shows the number of amino acids in low complexity divided by the total number of amino acids (=low complexity/length).

- *In the legend of Figure 1B, it might be helpful to say what the whiskers in the box plot represent.*

The boxplot is generated using R (boxplot.stats) with standard whiskers extending to +/-1.5* box height (i.e. total 3 times the interquartile range). We added this information to the legend.

- *What is the significance of font size in Figures 1C and S5B? These figures do not add much.*

We manually increased the font size of some terms based on biological significance and number of similar significant hits. Due to the large number of categories we cannot globally increase font size.

- *In a few instances, the exact p-value given is 0.0001. Is this really an exact p-value?*

Below $p=0.0001$ GraphPad Prism does not provide more accurate values.

- *In Figure S2, the legend should be clarified to say that the gray lines in the graph indicate whether a protein is significantly enriched/depleted in the cell type in question. The sentence, "Filled circles indicated that the protein was significantly enriched in both cell types" should be modified to clarify that whether a circle is filled in indicates whether the protein is significantly enriched/depleted in the other cell type.*

We adjusted the legend accordingly.

- *It might be helpful to explicitly state, for each DPR, how many proteins were significantly enriched in both cell types (as opposed to each cell type individually).*

The most interesting numbers are found in the text. Other counts can be deducted from Table S1.

- *In Figure S3, it looks as though RFP-NOP56 drives decreased cytoplasmic localization of GFP-(GR)149 (presumably due to increased nucleolar localization) in HEK293FT cells. Is this representative?*

We don't think NOP56 has a systematic effect on cytoplasmic distribution of GFP-(GR)₁₄₉.

- *It might be more helpful to the reader to provide the number of cells analyzed than the number of images analyzed.*

It's unclear to what figure this comment refers. The statistics are accurately described in all legends.

- *In the LC-MS/MS part of the methods section, the flow rate is described as 300 nL/mL.*

We corrected this to 300 nL/min.

- *In addition to providing the proportion of GFP-(GR)53-transduced neurons that had nucleolar GFP-(GR)53, it might be good to mention the number of independent samples analyzed.*

We analyzed 6 to 16 images and indicated this in the revised legend.

- *It might be helpful to show examples of the homogenous, ring-shaped, and granular distributions of fibrillarin.*

We think this is obvious from the description in the text.

- *When anti-DPR antibodies are used that recognize a specific methylation status, please specify this.*

Anti-GR clone 7H1 was raised against asymmetrically di-methylated GR-peptide and robustly detects also non-methylated and less efficiently symmetrically di-methylated GR (Schludi et al, Acta Neuropathol 2015). anti-PR 32B3 was raised against non-methylated PR-peptide. Arginine-methylation of poly-PR has not been reported so far. This information is now provided in the methods.

- *Clarify whether the "synthetic" genes encoding DPRs use alternative codons to prevent the formation of RNA foci.*
-

The synthetic DPR constructs use alternative codons in order to largely exclude RNA based toxicity and to focus on protein-based toxicity as mentioned in the original publication (May et al, Acta Neuropathol 2014).

1st Editorial Decision

23 April 2018

Thank you for submitting your revised manuscript entitled "Proteomics and C9orf72 neuropathology identify ribosomes as poly-GR/PR interactors driving toxicity". The manuscript was previously reviewed at a different journal and the referee reports have been transferred to Life Science Alliance. You provided a revised manuscript and a detailed point-by-point response to the reports obtained during peer-review elsewhere.

As outlined to you before submission to our journal, we expected that a revision based on the reports obtained elsewhere includes a full list of enrichment scores for GO terms (Ref#1, minor concern #6), tones-down some of your original conclusions (referee #3, first major point), and addresses the concern raised regarding statistics/quantifications (referee #3). Both the academic editor and I appreciate the full point-by-point response provided and the changes introduced in revision, addressing these requests. We are thus happy to accept your manuscript in principle for publication in Life Science Alliance.

2nd Editorial Decision

28 April 2018

Thank you for contributing your Research Article entitled "Proteomics and C9orf72 neuropathology identify ribosomes as poly-GR/PR interactors driving toxicity". It is a pleasure to let you know that your manuscript is now accepted for publication in Life Science Alliance. Congratulations on this interesting work.